# COVID-19: Pain Management in Patients with SARS-CoV-2 Infection—Molecular Mechanisms, Challenges, and Perspectives

**DOI:** 10.3390/brainsci10070465

**Published:** 2020-07-20

**Authors:** Sylwester Drożdżal, Jakub Rosik, Kacper Lechowicz, Filip Machaj, Bartosz Szostak, Paweł Majewski, Iwona Rotter, Katarzyna Kotfis

**Affiliations:** 1Department of Pharmacokinetics and Monitored Therapy, Pomeranian Medical University in Szczecin, 70-111 Szczecin, Poland; starkdrozd@wp.pl; 2Department of Physiology, Pomeranian Medical University in Szczecin, 70-111 Szczecin, Poland; jakubrosik97@gmail.com (J.R.); machajf@gmail.com (F.M.); bartszost1@gmail.com (B.S.); 3Department of Anesthesiology, Intensive Therapy and Acute Intoxications, Pomeranian Medical University in Szczecin, 70-111 Szczecin, Poland; kacper.lechowicz@gmail.com; 4Department of Anesthesiology and Intensive Therapy, Regional Specialist Hospital, 72-300 Gryfice, Department of Cardiac Surgery, Ceynowa Hospital, 84-200 Wejherowo, Poland; lek.majewski.pawel@gmail.com; 5Department of Medical Rehabilitation and Clinical Physiotherapy, Pomeranian Medical University in Szczecin, 71-210 Szczecin, Poland; iwona.rotter@pum.edu.pl

**Keywords:** coronavirus, neuropathy, myalgia, arthralgia, headache, opioids, NSAIDs, CPOT, BPS

## Abstract

Since the end of 2019, the whole world has been struggling with the pandemic of the new Severe Acute Respiratory Syndrome Coronavirus (SARS-CoV-2). Available evidence suggests that pain is a common symptom during Coronavirus Disease 2019 (COVID-19). According to the World Health Organization, many patients suffer from muscle pain (myalgia) and/or joint pain (arthralgia), sore throat and headache. The exact mechanisms of headache and myalgia during viral infection are still unknown. Moreover, many patients with respiratory failure get admitted to the intensive care unit (ICU) for ventilatory support. Pain in ICU patients can be associated with viral disease itself (myalgia, arthralgia, peripheral neuropathies), may be caused by continuous pain and discomfort associated with ICU treatment, intermittent procedural pain and chronic pain present before admission to the ICU. Undertreatment of pain, especially when sedation and neuromuscular blocking agents are used, prone positioning during mechanical ventilation or extracorporeal membrane oxygenation (ECMO) may trigger delirium and cause peripheral neuropathies. This narrative review summarizes current knowledge regarding challenges associated with pain assessment and management in COVID-19 patients. A structured prospective evaluation should be undertaken to analyze the probability, severity, sources and adequate treatment of pain in patients with COVID-19 infection.

## 1. Introduction

Since the end of 2019, the whole world has been struggling with the epidemic of the new Severe Acute Respiratory Syndrome Coronavirus (SARS-CoV-2), which was first detected in the Chinese province of Hubei. Not long after its outbreak, the Coronavirus Disease 2019 (COVID-19) was declared an epidemic by the World Health Organization (WHO) and has become a global health threat [1]. The novel virus is classified as a member of the Coronaviridae family, single-stranded RNA viruses [2,3]. In recent history, there have been recorded human infections with other viruses from this family: Severe Acute Respiratory Syndrome (SARS) caused by SARS-CoV-1 and Middle East Respiratory Syndrome (MERS) caused by MERS-CoV [4,5].

The main sources of infection spread are droplets and direct contact [6,7,8,9]. The estimated spread rate ranges from 2.2 to 3.58, while the average incubation time lasts about 5 days [10,11]. Those factors significantly impede the fight against the virus, with the lack of targeted treatment and vaccine further contributing to the spread of infection. The symptoms of infection are usually nonspecific, ranging from common cold-like to severe, and sometimes lethal, respiratory infection [7,8,11,12]. Studies suggest that, in many cases, the infection might be asymptomatic [7,8,12].

One of the most notable problems of severely ill patients with COVID-19 is comorbidity. It has been reported that 50% of Chinese study participants had 1 or more pre-existing medical conditions, mostly cardiovascular or cerebrovascular [13,14,15].

The problems associated with pain management during the current pandemic are multiple and, on one hand, they relate to the new onset of symptoms and, on the other, they are caused by problems related to restricted access to chronic pain services. There are many challenges associated with pain treatment during the COVID-19 pandemic that force both health care workers and patients to operate in drastically different conditions [16]. Available evidence and WHO reports suggest that pain is a common symptom during infection with SARS-CoV-2. These pain-related symptoms primarily include: muscle pain (myalgia) and/or joint pain (arthralgia)—14.8%—sore throat—13.9%—and headache—13.6% [17].

The isolation of the patient inevitably leads to greater stress and anxiety, which may lead to exacerbation of symptoms not only related to COVID-19, but also worse clinical condition, and further deterioration with a higher risk for delirium [18] and depression [19,20]. This may result in the worsening of the painful symptoms, which in untreated chronic pain conditions may lead to depression in 50% of patients [21]. Cancer patients are a special group exposed to pain and pain-related treatment problems during the current COVID-19 pandemic.

The main risk for these patients is limited access to healthcare facilities and services and the inability to get help at the right time [22]. It results from the fact that many patients cannot receive appropriate care and help from their relatives and friends who, because of the risk of epidemic spread, are forced to leave their loved ones to the in-hospital care and clinical personnel. This process excludes the family from the process of treatment, care, and support [16,23].

COVID-19 patients suffer not only from respiratory, circulatory, and gastrointestinal symptoms, but they also show neurological symptoms, including peripheral neuropathies. On the other hand, symptoms that could be related to COVID-19, such as dry cough, tiredness, nausea, and gastrointestinal could be worsened by using opioids and other medicines used for relieving neuropathic pain. Some kind of persistent pain like neck, back, orofacial, headache, and cervical/lumbar pain may worsen during COVID-19 and these patients should receive additional care and support as infection may increase their analgesic requirement [24].

## 2. Pathogenesis of Pain during Coronavirus Infection

In COVID-19, more than one-third of patients experience different neurological symptoms, which may involve the central nervous system (dizziness, headache, impaired consciousness, acute cerebrovascular disease, ataxia, and epilepsy), the peripheral nervous system (taste impairment, smell impairment, vision impairment, and neuralgia) and skeletal muscular damage [25,26]. Mao et al. reported peripheral nervous system (PNS) effects in their study presenting in the form of dysgeusia (5.6%), dysosmia (5.1%), visual disturbances (1.4%), and neuralgia (2.3%) [26]. Abdelnour et al. reported a case of peripheral neuropathy manifesting before the onset of the typical flu-like symptoms of the novel COVID-19 infection [27]. In this case, the patient had no flu-like symptoms until day seven of symptom onset had distal lower limb weakness and hyporeflexia without back pain or sensory level, suggestive of peripheral neuropathy [27].

Sheraton et al. hypothesized that CNS symptoms may occur due to the inflammatory mechanisms and PNS due to immune-mediated processes [28], but more research is needed to explain SARS-CoV-2 related neuropathy [29]. Meanwhile, the number of reports on patients suffering from COVID-19 with neurological symptoms is growing and the role of the SARS-CoV-2 role in the neuropathogenic invasion remains unclear and needs further studies [29,30]. Helms et al. showed that patients with COVID-19 had neurological symptoms such as perfusion abnormalities, confusion, agitation and ischemic stroke. There is no data to determine whether the occurrence of these disorders is a characteristic or coincidence [30]. Su et al. suggest that central pain could be induced through the ACE2-positive cells in the human spinal dorsal horn via the decrease of functional ACE2 (Angiotensin-converting enzyme 2), which then results in the accumulation of Ang. II (Angiotensin II) and the decrease of Ang. (1–7) (Angiotensin 1–7) [31]. The pain induced by COVID-19 infection could result from the effect of spinal ACE2 on pain sensation and the direct or indirect tissue damage, but the ACE2 role in the transmission and management of pain in infected patients needs further investigation [31].

One of the most common causes of pain in COVID-19 infection is the associated muscle pain. Multiple studies have shown that myalgia is one of the most common symptoms at onset, seen in nearly 36% of patients [32]. Myalgia during viral infection is most commonly mediated by Interleukin-6 (IL-6), whose upregulation causes muscle and joint pain [33]. It is believed that myalgia in COVID-19 patients might reflect the generalized inflammation and cytokine response [34]. As SARS-CoV-2 induces a strong inflammatory response, elevated cytokine levels (IL-6, IL-10, and TNF α) are present, especially in patients with a moderate or severe disease course [35,36].

Inflammasomes are a crucial part of the inflammatory cascade [37,38]. They are engaged in the production of proinflammatory cytokines [37]. Microglia, astroglia and neurons are numbered amongst cells expressing inflammasomes [39,40,41]. Specific inflammatory ligands (pathogen-associated molecular patterns and damage-associated molecular patterns) participate in the activation of inflammasomes [42,43]. The former ones are particles of pathogens like nucleic acid or lipopolysaccharides, the latter ones are released from distressed cells in the central nervous system. However, the SARS-CoV-2 presence in brain inducing inflammasomes is unlikely because the brain sections of patients who died due to COVID-19 did not show any cytoplasmic viral staining or encephalitis [44]. Therefore, pain pathogenesis must involve the inflammatory reaction [37]. Microglial cells participating in this process might promote either regeneration or toxic neuroinflammation [45], depending on many factors, such as gonadal steroid hormones [46,47]. It might lead to differences between genders in the clinical presentation [46]. Microglia are implicated in the spinal cord [37,48]. Unfortunately, the link between analgesics and inflammasomes’ role in pain transmission is not well elucidated [37].

The exact mechanisms of headache and myalgia during viral infection are still uncertain [49,50]. Figure 1 presents the most likely pathomechanism of those common viral infection symptoms. During the COVID-19 pandemic, patients with neurologic manifestations and suffering from different types of pain cannot be ignored. The neurologic manifestation of the SARS-CoV-2 infection seems to be underestimated and they should be taken into account with significant care to set appropriate diagnosis and prevention of the transmission of the infection. Especially for patients with COVID-19 neurological symptoms, these manifestations may contribute to the severity of the infection and lead to rapid deterioration and death [26]. Significant neurological system involvement is an additional reason to undertake further research and studies regarding the diagnostics, mechanisms, and therapeutic options in COVID-19 [51].

## 3. Difficulties Associated with Reliable Pain Assessment during COVID-19

Many patients with SARS-CoV-2 infection will suffer from severe pain and require reliable pain assessment to provide adequate analgesia, often with multiple drugs, including opioids, non-steroidal inflammatory drugs or analgosedation [52]. The golden standard for the assessment of pain are tools based on patient self-assessment, e.g., visual analogue scale (VAS) or numerical assessment scale (NRS), which assume patient–physician cooperation [53]. Assessing the intensity of pain in intubated, non-verbal patients with respiratory failure who are admitted to the intensive care unit (ICU), mechanically ventilated and sedated, remains a constant challenge for ICU clinicians [53,54,55,56]. 

Pain in ICU patients can be divided into four categories: acute pain associated with the disease, continuous pain/discomfort associated with ICU treatment, intermittent procedural pain and chronic pain present before admission to the ICU [53]. Daily care and medical interventions in the ICU can also be a potential source of pain, so it is very important to use simple tools to monitor it. Undertreatment of pain, especially when using neuromuscular blocking agents, prone positioning or extracorporeal membrane oxygenation (ECMO) may trigger delirium [18,57] and cause persistent neuropathies [54]. Therefore, it is recommended that, in patients who are unable to self-report pain, behavioral pain assessment scales should be used, namely the Behavioral Pain Scale (BPS) and the Critical Care Pain Observation Tool (CPOT) [55,56]. The CPOT scale was designed for the critical detection of pain in sick patients and includes four behavioral categories—facial expressions, body movements, muscle tone, susceptibility with a fan (for intubated patients) or verbalization (for extubated patients). Each category is scored on a 0–2 scale (0–8 points in total) [55,58]. BPS was developed by Payen et al. to assess pain in mechanically ventilated unconscious patients. The scale is based on three types (ranges) of behavior: facial expressions, upper limb movements and ventilation compatibility [56,59]. All these elements of pain assessment in a patient admitted to the ICU, including patients infected with COVID-19, can be helpful in selecting appropriate treatment.

## 4. Somatic Pain Treatment in SARS-CoV-2 Infections

There are currently no clinical trials or specific guidelines regarding the topic of pain management in COVID-19 patients [60,61]; therefore, the aim of this narrative review is to discuss the problems associated with pain treatment during the COVID-19 pandemic. Optimal pain management may be extremely challenging in mechanically ventilated COVID-19 patients who are often deeply sedated and receive neuromuscular blocking medications. Both the nursing and physiotherapy teams may be spending less time at the patients’ side making pain management suboptimal and extended periods of high-dose intravenous opioid infusions may be inevitable [62].

### 4.1. NSAIDs and Non-Opioid Analgesics

Nonsteroidal Anti-Inflammatory Drugs (NSAIDs) are one of the most widely used drugs worldwide. The popularity of NSAIDs is the result of their easy accessibility and efficacy as anti-inflammatory and analgesic agents. They can be divided based on their chemical structure or ability to selectively inhibit cyclooxygenase isoenzymes [63]. The chemical classification of NSAIDs is presented in Figure 2. NSAIDs influence the arachidonic acid cascade and prostaglandins biosynthesis through the inhibition of cyclooxygenase (COX) [64]—Figure 3.

The COX enzyme was firstly described in 1976, and later the existence of at least two isoforms of the enzyme (COX-1 and COX-2) was established [65]. The constitutive expression in various tissues is characteristic for the COX-1 isoenzyme. Its activity is responsible for maintaining homeostasis, but particularly for gastrointestinal mucosal cytoprotection, the regulation of renal blood flow, platelet aggregation, and endothelial functioning. COX-2 expression is highly restricted in a physiological state; however, it is constitutively active in the brain, spinal cord, kidneys, testes, and bronchial epithelium. Not only inflammatory mediators, but also hormones secreted by the ovaries, uterus, and fetal membranes can greatly induce COX-2 expression.

Furthermore, this isoenzyme influences the menstrual cycle and embryo implantation. The discovery of the COX-2 isoenzyme further led to the development of its selective inhibitors, as the main activity of this enzyme is linked with inflammation. This approach was established to reduce the renal and gastrointestinal adverse effects associated with the blockade of COX-1 [66].

Recently, concerns about the possible higher frequency of adverse effects and exacerbation of symptoms of viral respiratory tract infections, such as COVID-19, in patients treated with NSAIDs have been raised [67]. However, according to the WHO (as of 19th of April 2020), there is no evidence for the aforementioned hypothesis [68]. The WHO has evaluated 73 studies, conducted on adults and children treated with NSAIDs for respiratory tract infection. However, in none of the studies was the infection caused by COVID-19, SARS or MERS [68]. 

The number of infected by SARS-CoV-2 is constantly increasing worldwide. The supportive therapy is the main treatment regimen for patients with mild and moderate clinical symptoms of COVID-19, and the widely available NSAIDs, such as ibuprofen are commonly used. The controversies regarding the safety of ibuprofen in COVID-19 have emerged after a report to public media was made by a French infectious disease specialist, who observed the decompensation and development of severe symptoms in an early stage of infection in four children after administration of ibuprofen [69]. The report was firstly confirmed by the French Minister of Health and the WHO. However, having reviewed the available data, the WHO published their recommendations, underlining no evidence of COVID-19 patient decompensation after the usage of NSAIDs [67]. Nonetheless, the previous study by Kotsiou et al. reported that pre-hospital usage of NSAIDs in the treatment of symptoms of community-acquired pneumonia is connected with the exacerbation of pneumonia, prolonged hospitalization, and more severe pleural effusions [70]. Several studies suggest that this effect should be considered in COVID-19 treatment [71,72].

In conclusion, no study to date has reported higher mortality of COVID-19 patients treated with ibuprofen [73]. Despite this, in doubtful cases, paracetamol or metamizole should be used instead of ibuprofen, especially in over the counter (OTC) usage [74]. 

There are, however, some cases where the use of paracetamol might not be beneficial, due to its pharmacokinetic properties. As paracetamol is metabolized mainly by two cytochrome P450 isoenzymes—CYP1A2 and CYP2E1—cytokine storm associated with viral infection can potentially disrupt the metabolic functioning of cytochrome P450 [75]. This disturbance could paradoxically escalate the risk of adverse effects in patients treated with paracetamol. As far as metamizole is concerned, it does not interact with therapeutic agents, commonly used in SARS-CoV-2 therapy [76]. Moreover, it affects SARS-CoV-2 main protease (Mpro), curbing its transcription and replication [77]. However, the role of metamizole in infection symptoms control is questionable because of its association with agranulocytosis [78]—the adverse effect that could lead to a sudden worsening of COVID-19 patient condition.

Overall, both the EMA (European Medicines Agency) and the WHO do not recommend altering or discontinuing NSAID therapy in patients using them in chronic treatment, with suspected or diagnosed SARS-CoV-2 infection. At this stage, there is a lack of evidence suggesting that therapeutic regimens should be changed to other non-opioid analgesics [79].

It is also worth remembering that NSAIDs could interfere with the inhibition of platelet COX-1 by aspirin. The number of patients using acetylsalicylic acidin the anti-platelet dosing range is still increasing and any influence on their therapy could result in an increased risk of cardiovascular events. Cardiovascular or cerebrovascular comorbidities of COVID-19 patients preclude the use of NSAIDs. Moreover, heart failure and substantial cardiac damage, which also preclude NSAIDs administration, lead to around 12% of deaths amongst COVID-19 patients [13,14,15].

### 4.2. Opioids

Although the use of opioids for pain management inside and outside of an ICU is inevitable, problems associated with their use must be acknowledged. The lack of clinical trials conducted on patients infected with CoVs (SARS, MERS or COVID-19) makes it impossible to unambiguously estimate their usefulness during the COVID-19 pandemic. 

Opioids have been proven to reduce dyspnea amongst patients with lung diseases, such as chronic obstructive pulmonary disease (COPD), or lung cancer [80,81,82]. Ekstrom et al. in their meta-analysis found that small doses of opioids are safe and efficacious in reducing shortness of breath [80]. However, they do not improve exertion capacity in advanced COPD. Woodcock et al. and Light et al. found that oral administration of single-dose morphine significantly decreased tidal volume and respiratory frequency during exercises in isotime and reduced nervous inspiratory drive to the larynx and diaphragm [83,84]. These results are coherent with opioids influence on the chemoreflex that influences the respiratory drive [85]. Opioids have been found to increase both mortality and the risk of adverse clinical events amongst patients with severe pulmonary disease [82,86,87]. Moreover, morphine and diamorphine are used to reduce respiratory disturbances [88].

Yamamoto et al. verified the effects of oxycodone intravenous administration. It was found to reduce dyspnea in almost 70% of patients, without causing any significant adverse effects [89]. Fentanyl, in comparison with oxycodone and morphine, significantly reduced brain oxygen supply [85]. Similar to other opioids, its use was associated with not only decreases in respiratory rate, mediated by μ receptor agonism, but also reduced tidal volume [88]. Respiratory depression induced by fentanyl is similar to that induced by oxycodone and subsides much sooner than with morphine or buprenorphine use [90]. Differences in the pharmacodynamic properties of fentanyl lead to a significantly higher risk of overdose than other opioids used in medicine or heroine [91]. Moreover, fentanyl overdose might lead to difficulties in intubation [83,92].

Another property of opioids is the induction of immunosuppression. In an animal model, Flores et al. found that morphine induces adrenal-dependent lymphopenia and reduces the response to mitogenic stimulation (dose = 10 mg/kg) by nearly 70% [91,92]. The suppression of immunity after fentanyl administration is dose dependent [52]. With regards to a decrease in NK cells’ activity [93,94,95,96] and the concentration of proinflammatory cytokines, a change in Il-1 [97] and Il-6 [97] are the most notable characteristics [98]. The immunosuppressing properties of fentanyl over the course of therapy decrease sooner than during morphine treatment [52]. Meanwhile, oxycodone immunosuppressing properties are not totally understood. The results of preclinical studies do not offer a coherent argument regarding its effect on lymphocyte proliferation [83,84]. However, multiple dissertations suggest that oxycodone has a higher safety profile than morphine or fentanyl [83,84,85,88,91]. The quality of clinical data concerning this topic is poor. Hernandez et al. suggest that oxycodone immunosuppressing effect subsides sooner (after around 6 h) than after morphine administration [92]. Wiese et al. have found that the immunosuppressing attributes of opioids influence the frequency of infections [99]. The relative risk (RR) of oxycodone treatment comparing to morphine was 0.73 [99]. Other opioids, tramadol and buprenorphine seem to be a clinically superior choice. Neither of them has immunosuppressive properties, so, in theory, they do not prolong viral shedding [100,101]. In addition, buprenorphine is safe in multiorgan failure and has a ceiling effect for respiratory depression [100,101].

### 4.3. Corticosteroids

Corticosteroids were widely used in clinics during the SARS-CoV outbreak and several positive effects of their use were noted, which is attributed mainly to their ability to modulate the inflammatory response. Multiple studies were established in that time and reached conclusions that steroids are efficacious in decreasing the extent of immunopathological damage. However, the side effects indivertibly associated with their long-term use dissuaded physicians from prolonged therapy, fearing the rebound effect of the infection, and consequentially the development of adverse effects, such as acute respiratory distress syndrome. In one of the randomized clinical trials, where viral load was measured in regular time periods in intubated patients with SARS-CoV, a higher concentration of viral RNA was noted during weeks 2 and 3 of infection in those treated with steroids in comparison with placebo [102].

One of the studies using a mice model identified the N-protein of SARS-CoV as a factor responsible for triggering the pulmonary inflammatory response and acute lung injury, which was associated with an increase and imbalance of proinflammatory cytokines. Dexamethasone was successful in alleviating this response in mice [103]. Similarly, animal studies involving swine infected with CoV provided further evidence that one or two doses of corticosteroid during the acute phase of the infection can successfully ameliorate the early inflammatory response; however, their prolonged use might promote replication of the virus [104]. 

Further observational studies aiming to evaluate the efficacy of agents used in SARS reported no survival benefit and possible harms (mainly delayed viral clearance) in patients treated with corticosteroids [105]. A study involving patients receiving corticosteroids in MERS infection, when adjusted for time-varying confounders, found no effect of corticosteroids on mortality. Moreover, the treatment delayed lower respiratory tract viral clearance [106]. Based on these findings, the current WHO guidelines for the management of severe acute respiratory infection with SARS-CoV-2 do not recommend routine corticosteroid use [12]. It is advised to weigh risk against benefit in individual patients, with the use of steroids being justified for the treatment of other concurrent conditions, such as asthma or COPD exacerbation, or septic shock. As far as the treatment of sepsis in COVID-19 is concerned, recent guidelines recommend conditional use of steroids for all patients with sepsis and septic shock, on the condition that the potential reduction in mortality outweighs prolonged coronavirus shedding. In those patients, glycemia, natremia, and kalemia must be monitored [12].

## 5. Neuropathic Pain Treatment in SARS-CoV-2 Infection

### Gabapentinoids

Peripheral nervous system involvement, including painful neuropathies, was reported in many patients with SARS-CoV-1 and now with SARS-CoV-2 infection [26,27,29,107]. This may be a consequence of either viral invasion of the peripheral nerves (neurotropism) [108] or prolonged immobility during severe illness [62], or both. Peripheral neuropathies are prevalent in COVID-19 patients and may require an addition of gabapentinoids to the pain treatment regime [62]. The gabapentinoids (gabapentin and pregabalin) are commonly used in pain management in adults, but there is a scarcity of research on their effectiveness in children [109]. Gabapentin and pregabalin are calcium channel α2-δ ligands commonly used in the treatment neuropathic pain. While the numbers needed to treat (NNT) for 50% pain relief for these therapeutic agents are similar (7.2—gabapentin; 7.7—pregabalin) [110,111] pregabalin acts quicker than gabapentin [112]. They are usually well tolerated [113] and characterized by similar adverse effects [110,111,112,114]. Their potency in the therapy of neuropathy acquired due to SARS-CoV-2 is hard to predict due to a lack of dissertations on gabapentinoids in CoV infections. Currently, there are no clinical trials exploring this topic. However, calcium channel ligands reduce respiratory drive; therefore, combined therapy with opioids might be potentially hazardous and the use of duloxetine in such cases ought to be thoroughly considered [115].

## 6. Summary

Apart from pandemic control and the prevention of the spread of COVID-19, medical staff worldwide try to improve patient care, including the quality of pain treatment. There are reports of a significantly higher use of opioids because of sedation requirements during respiratory failure caused by SARS-CoV-2, which highlights the importance of undertaking a study aiming to determine efficacious and safe procedures of pain management in patients with COVID-19. Apart from pain symptoms caused by the virus, including myalgia, arthralgia, sore throat, headache and peripheral neuropathies, problems associated with during ICU treatment (procedural pain, prolonged mechanical ventilation, muscle wasting, immobility during prone positioning) may arise. COVID-19, despite its prevalence, is virtually unknown. For the treatment of pain, each patient requires an individual approach based on available knowledge and, more importantly, the patient’s condition and comorbidities. The information provided is a cross-section of the available knowledge aimed at improving the patient’s clinical condition. A structured prospective evaluation should be undertaken to analyze the probability, severity, sources and adequate treatment of pain in patients with COVID-19 infection and those suffering due to an unavailability of pain services during the COVID-19 pandemic.

## Figures and Tables

**Figure 1 brainsci-10-00465-f001:**
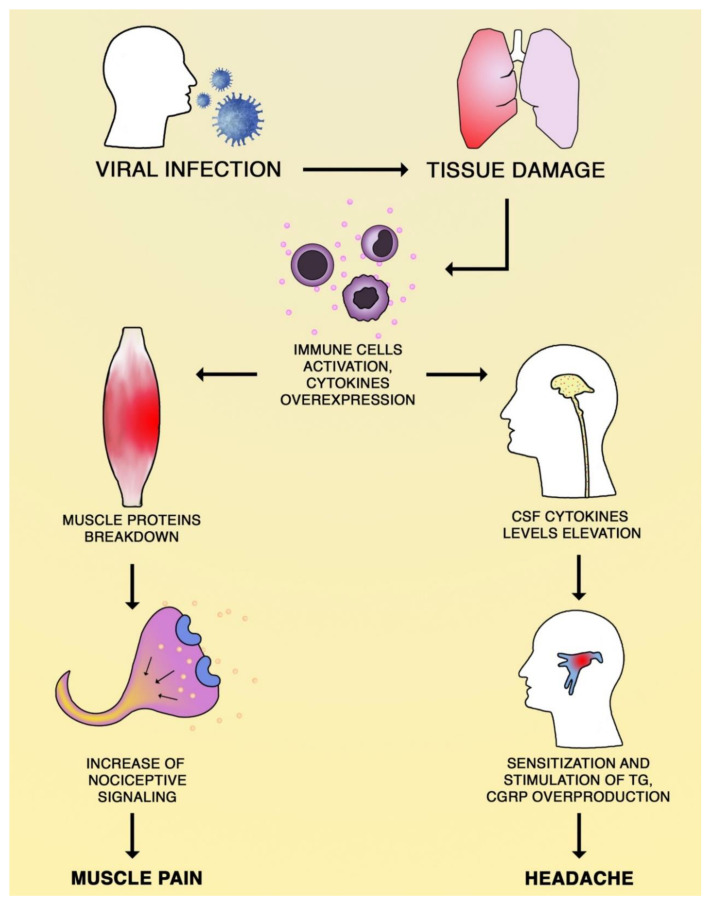
Putative mechanisms of myalgia and headache during viral infection. The specific mechanism of headache during infection remains unclear. The assumed pathomechanisms involve the overexpression of proinflammatory cytokines, such as TNF-α and PGE2, in the cerebrospinal fluid (CSF), which sensitize and stimulate trigeminal ganglia (TG) neurons to produce calcitonin gene-related peptide (CGRP). CGRP has a crucial role in the pathogenesis of migraine, influencing arteries’ dilatation and possibly to direct a nociceptive transmission. The myalgia during viral infection is believed to be the effect of proinflammatory cytokines influence on muscle tissue. TNF-α is responsible for the intensified breakdown of muscle proteins and PGE2 could increase the nociceptive signaling.

**Figure 2 brainsci-10-00465-f002:**
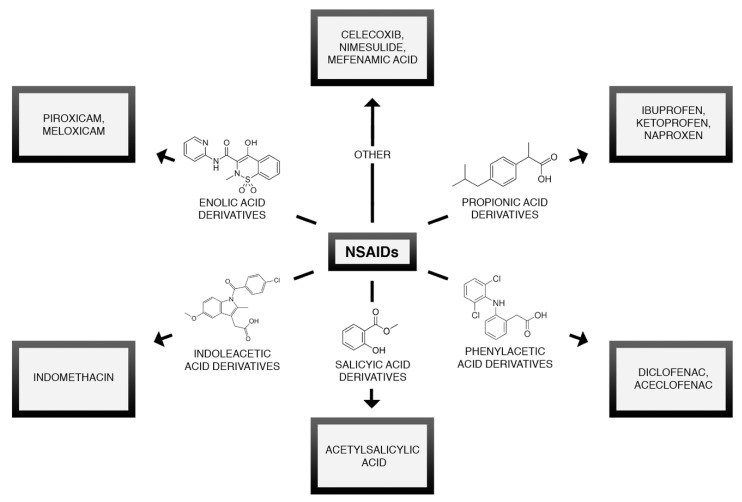
Chemical classification of NSAIDs.

**Figure 3 brainsci-10-00465-f003:**
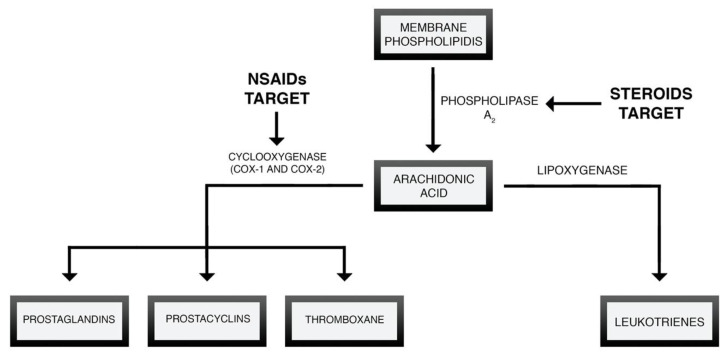
Initial stage of the cascade is mediated by phospholipase A2, whose activity leads to the release of arachidonic acid from membrane phospholipids. Arachidonic acid can be further transformed by cyclooxygenase (COX) into prostanoids (prostaglandins, prostacyclins and thromboxane) or by lipoxygenase (LOX) into leukotrienes.

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
