# Peer review of "COVID-19: Pain Management in Patients with SARS-CoV-2 Infection—Molecular Mechanisms, Challenges, and Perspectives"

_brainsci, 2020, doi:10.3390/brainsci10070465_

Round 1

Reviewer 1 Report

This review summarizes the current knowledge related to the different drugs used as pain relievers for the treatment of pain in COVID-19 patients.
The review reported a collection of studies evaluating different classes of NSAIDs and non-opioid analgesics, corticosteroids and opioids.
A short paragraph on the use of gabapentinoids in the treatment of neuropathic pain is also reported.
The main concern is that, after the description of all analgesic classes, no critical analyzes and perspectives are reported in the summary as conclusions.In Figure 3 the authors summarized the mechanism and pathway of inflammation. But if they consider useful to sum up this well known cascade, the picture is unclear, since enzymes are not clearly distinguished from substrates. So, this picture must be replaced with a more clear scheme.As a minor concern, the document contains too many abbreviations and many are not explained in the text.
There are also several punctuation errors.

Author Response

Dear Reviewer,

Thank you for your effort and helpful comments regarding our manuscript.

Reviewer 1

Comments and Suggestions for Authors

This review summarizes the current knowledge related to the different drugs used as pain relievers for the treatment of pain in COVID-19 patients.

The review reported a collection of studies evaluating different classes of NSAIDs and non-opioid analgesics, corticosteroids and opioids.

A short paragraph on the use of gabapentinoids in the treatment of neuropathic pain is also reported.

The main concern is that, after the description of all analgesic classes, no critical analyzes and perspectives are reported in the summary as conclusions.

  • Response: Thank you for these helpful suggestions. COVID-19 is a disease that is very much unknown. In our article, we tried to present the possibilities of pain treatment based on facts from the available literature. The approach to each patient requires action directed individually, depending on the patient's condition and associated diseases. We have added a few fragments to better explain the treatment options, and a short note in the summary.

The following changes have been made:

  • “One of the most notable problems of patients severely ill with COVID-19 is comorbidity. 50% of Chinese study participants had 1 or more pre-existing medical condition, mostly cardiovascular or cerebrovascular13–15.” (lines 54-56)
  • „Cardiovascular or cerebrovascular comorbidities of COVID-19 patients preclude the use of NSAIDs. Moreover, heart failure and substantial cardiac damage, which also preclude NSAIDs administration, lead to around 12% of deaths amongst COVID-19 patients13–15.” (lines 248-251)
  • „Opioids have been found to increase both mortality and the risk of adverse clinical events amongst patients with severe pulmonary disease82,86,87.” (lines 265-266)
  • “Other opioids, tramadol and buprenorphine seem to be a clinically superior choice. Neither of them has immunosuppressive properties, so in theory, they do not prolong viral shedding100,101. In addition, buprenorphine is safe in multiorgan failure and has a ceiling effect for respiratory depression100,101.” (lines 290-292)
  • However, calcium channel ligands reduce respiratory drive, therefore combined therapy with opioids might be especially hazardous and the use of duloxetine in such cases ought to be thoroughly considered115. (lines 338-340)
  • COVID-19, despite its prevalence, is virtually unknown. For the treatment of pain, each patient requires an individual approach based on available knowledge and, more importantly, the patient's condition and comorbidities. The information provided is a cross-section of the available knowledge aimed at improving the patient's clinical condition. (Summary, lines 349-353)

In Figure 3 the authors summarized the mechanism and pathway of inflammation. But if they consider useful to sum up this well-known cascade, the picture is unclear, since enzymes are not clearly distinguished from substrates. So, this picture must be replaced with a more clear scheme.

  • Response: Thank you for these suggestions. We’ve changed Figure 3 and I hope it is clearer.

As a minor concern, the document contains too many abbreviations and many are not explained in the text. There are also several punctuation errors.

  • Response: Thank you for these suggestions. All spelling and grammatical errors have been corrected by a native speaker. Thank you for pointing this out. We have developed some abbreviations to make them more readable in the text. We've also added explanations for the others.

With best regards

Katarzyna Kotfis

Reviewer 2 Report

Critique

  • This is a well written paper in very timely
  • The depth of the discussion is very good
  • The papers well-organized
  • There are several points I would like to make regarding the discussion of treatment
  • Patients who are extremely sick with COVID19 frequently have comorbidities. In the experience in China nearly 50% had 1 a more existing medical conditions mostly cardiovascular and cerebrovascular which would preclude the use of NSAIDS. In addition, nearly 12% died from substantial cardiac damage and heart failure which would preclude the use of NSAIDS1-3
  • Recent studies involving patients with severe pulmonary disease have found that the use of opioids increases the risk of adverse clinical events including increase mortality4-6
  • The choice of opioids may make a significant difference. Tramadol and buprenorphine are not immunosuppressive (if immunosuppression prolongs viral shedding as seen with corticosteroids). Buprenorphine has a ceiling on respiratory depression and is safe in multiorgan failure. I agree with the authors in regard to fentanyl7-9.
  • Gabapentin and pregabalin cause a reduction in respiratory drive which is compounded with the use of opioids. In this situation duloxetine may be a reasonable drug to use in people with respiratory embarrassment10.
  • 1.            Kishor K, Marwah R, Anantharaj A, Kalra S. Cardiovigilance in COVID 19. J Pak Med Assoc 2020;70(Suppl 3):S77-S80.
  • 2.            Kaluzna-Oleksy M, Gackowski A, Jankowska EA, et al. The patient with heart failure in the face of the coronavirus disease 2019 pandemic: an expert opinion of the Heart Failure Working Group of the Polish Cardiac Society. Kardiol Pol 2020;78:618-31.
  • 3.            Hulot JS. COVID-19 in patients with cardiovascular diseases. Arch Cardiovasc Dis 2020;113:225-6.
  • 4.            Vozoris NT, Pequeno P, Li P, Austin PC, O'Donnell DE, Gershon AS. Predictors of Opioid-related Adverse Pulmonary Events among Older Adults with COPD. Ann Am Thorac Soc 2020.
  • 5.            Vozoris NT. Opioid utility for dyspnea in chronic obstructive pulmonary disease: a complicated and controversial story. Ann Palliat Med 2020;9:571-8.
  • 6.            Vozoris NT, Wang X, Fischer HD, et al. Incident opioid drug use and adverse respiratory outcomes among older adults with COPD. Eur Respir J 2016;48:683-93.
  • 7.            Davis MP, Pasternak G, Behm B. Treating Chronic Pain: An Overview of Clinical Studies Centered on the Buprenorphine Option. Drugs 2018;78:1211-28.
  • 8.            Davis MP. Twelve reasons for considering buprenorphine as a frontline analgesic in the management of pain. J Support Oncol 2012;10:209-19.
  • 9.            Davis MP, Behm B. Reasons to avoid fentanyl. Ann Palliat Med 2020;9:611-24.
  • 10.          Gomes T, Juurlink DN, Antoniou T, Mamdani MM, Paterson JM, van den Brink W. Gabapentin, opioids, and the risk of opioid-related death: A population-based nested case-control study. PLoS Med 2017;14:e1002396.

Author Response

Reviewer 2

Comments and Suggestions for Authors

This is a well written paper in very timely

The depth of the discussion is very good

The paper is well-organized

  • Response: Thanks for your kind opinion regarding our manuscript.

There are several points I would like to make regarding the discussion of treatment.

  • Response: Thank you for these suggestions. It is with great pleasure that we have added these parts to the discussion. We believe that they will significantly increase the quality of the article.

Here are the fragments we have changed:

“Patients who are extremely sick with COVID19 frequently have comorbidities. In the experience in China nearly 50% had 1 a more existing medical conditions mostly cardiovascular and cerebrovascular which would preclude the use of NSAIDS.”

  • Response: “One of the most notable problems of patients severely ill with COVID-19 is comorbidity. 50% of Chinese study participants had 1 or more pre-existing medical condition, mostly cardiovascular or cerebrovascular13–15.” (lines 54-56)

“In addition, nearly 12% died from substantial cardiac damage and heart failure which would preclude the use of NSAIDS1-3”

  • Response: „Cardiovascular or cerebrovascular comorbidities of COVID-19 patients preclude the use of NSAIDs. Moreover, heart failure and substantial cardiac damage, which also preclude NSAIDs administration, lead to around 12% of deaths amongst COVID-19 patients13–15.” (lines 248-251)

Recent studies involving patients with severe pulmonary disease have found that the use of opioids increases the risk of adverse clinical events including increase mortality4-6

  • Response: „Opioids have been found to increase both mortality and the risk of adverse clinical events amongst patients with severe pulmonary disease82,86,87.” (lines 265-266)

The choice of opioids may make a significant difference. Tramadol and buprenorphine are not immunosuppressive (if immunosuppression prolongs viral shedding as seen with corticosteroids). Buprenorphine has a ceiling on respiratory depression and is safe in multiorgan failure. I agree with the authors in regard to fentanyl7-9.

  • Response: “Other opioids, tramadol and buprenorphine seem to be a clinically superior choice. Neither of them has immunosuppressive properties, so in theory, they do not prolong viral shedding100,101. In addition, buprenorphine is safe in multiorgan failure and has a ceiling effect for respiratory depression100,101.” (lines 290-292)

Gabapentin and pregabalin cause a reduction in respiratory drive which is compounded with the use of opioids. In this situation duloxetine may be a reasonable drug to use in people with respiratory embarrassment10.

  • Response: “However, calcium channel ligands reduce respiratory drive, therefore combined therapy with opioids might be especially hazardous and the use of duloxetine in such cases ought to be thoroughly considered115.” (lines 338-340)

With best regards

Katarzyna Kotfis

Round 2

Reviewer 1 Report

The authors tried to better explain the unclear points in this version.